# Influence of Nitrogen on Grapevine Susceptibility to Downy Mildew

**DOI:** 10.3390/plants12020263

**Published:** 2023-01-06

**Authors:** Demetrio Marcianò, Valentina Ricciardi, Giuliana Maddalena, Annamaria Massafra, Elena Marone Fassolo, Simona Masiero, Piero Attilio Bianco, Osvaldo Failla, Gabriella De Lorenzis, Silvia Laura Toffolatti

**Affiliations:** 1Department of Agricultural and Environmental Sciences, University of Milan, 20133 Milano, Italy; 2Department of Biosciences, University of Milan, 20133 Milano, Italy

**Keywords:** oomycete, *Vitis vinifera*, nitrogen fertilization, plant pathogen interaction, susceptibility genes, resistance genes

## Abstract

Downy mildew, caused by the obligate parasite *Plasmopara viticola*, is one of the most important threats to viticulture. The exploitation of resistant and susceptibility traits of grapevine is one of the most promising ways to increase the sustainability of disease management. Nitrogen (N) fertilization is known for influencing disease severity in the open field, but no information is available on its effect on plant-pathogen interaction. A previous RNAseq study showed that several genes of N metabolism are differentially regulated in grapevine upon *P. viticola* inoculation, and could be involved in susceptibility or resistance to the pathogen. The aim of this study was to evaluate if N fertilization influences: (i) the foliar leaf content and photosynthetic activity of the plant, (ii) *P. viticola* infectivity, and (iii) the expression of the candidate susceptibility/resistance genes. Results showed that N level positively correlated with *P. viticola* infectivity, confirming that particular attention should be taken in vineyard to the fertilization, but did not influence the expression of the candidate genes. Therefore, these genes are manipulated by the pathogen and can be exploited for developing new, environmentally friendly disease management tools, such as dsRNAs, to silence the susceptibility genes or breeding for resistance.

## 1. Introduction

Nitrogen (N) is the nutrient element applied in the largest quantity in cultivated soils [1,2]. In plants, the production of amino acids, nucleic acids, chlorophylls, and hormones, among other cellular components, requires nitrogen [2]. Nitrogen promotes plant growth, delays maturity, and affects cell size and cell wall thickness. It is also involved in plant resistance to pathogens, since it is essential for the production of antimicrobial compounds, such as phytoalexins and phenolics. Nitrogen stimulates vegetative growth, vine productivity, and grape quality [3]. In viticultural soils, the availability of nitrogen is usually insufficient to optimize the growth and productivity of crop. To overcome this, farmers use a huge number of fertilizers. However, the increase in nitrogen availability may have consequences on the susceptibility of the vine to fungal diseases, including downy mildew (caused by the oomycete *Plasmopara viticola*), for indirect effects related to the modification of canopy microclimate and to the persistence in the growth of main and lateral shoots, whilst the possible direct effects of the nitrogen nutritional status on the interaction mechanisms of parasite host are controversial [4].

The management of grapevine downy mildew is based on the integration of all available tools (cultivation of resistant cultivars, adoption of agronomic practices, and the use of resistance inducers and/or plant protection products with antifungal activity) finalized to reduce yield losses. Agronomic practices have an important role in supporting the cultivation of resistant grapes and chemical control. They aim to reduce the conditions that favor pathogen infections, such as excessive canopy growth induced by vigorous rootstocks and excessive fertilization (https://www.agric.wa.gov.au/table-grapes/downy-mildew-grapevines?nopaging=1, accessed on 20 November 2022). Moderate N fertilization is always recommended [5], since excessive nitrogen fertilization triggers uncontrolled shoot growth and dense canopies, favoring high humidity as well as fungal infections [6]. Moreover, cells exposed to nitrogen over-fertilization develop thinner and weakened cell walls, increasing disease risk [7]. A multifactor analysis highlighted that, among the others, leaf nitrogen content has an influence on the downy mildew severity in grapevine (*Vitis vinifera*) [8]. Even if nitrogen fertilization is known for influencing the disease severity of *P. viticola* in the open field, no information is available on the specific mechanisms underlying its role in the plant-pathogen interaction. This is because the disease severity is commonly evaluated in vineyard where climatic conditions, pathogen genetics, and host phenology vary greatly [9]. Furthermore, studies on *P. viticola* are difficult, since it is a biotrophic obligate plant pathogen [10].

Metabolic reprogramming is a general feature of plant–pathogen interactions [11]: it is necessary for the plant to synthesize defense metabolites and for the pathogen to acquire nutrients (C and N) from its host. The expression of N genes, in particular, is commonly affected by pathogen infection. A recent comparative transcriptome analysis, where the transcriptome of inoculated and non-inoculated grapevine cultivars (cv) with different resistance levels towards *P. viticola*, showed that different plant genes involved in N metabolism are differentially modulated by the pathogen [12]. In detail, genes encoding a glutamine synthetase (*VvGS*) and a high-affinity nitrate transporter (*VvNRT3.1*) were upregulated in the resistant cv Bianca and not differentially expressed or downregulated in the susceptible cv Pinot noir, while an asparagine synthetase (*VvAS1*) was highly upregulated in the susceptible cv Pinot noir and downregulated in the resistant cv Bianca. These genes are potentially involved in the mechanisms of grapevine resistance (*VvGS* and *VvNRT3.1*) or susceptibility (*VvAS1*) to *P. viticola*, and could be exploited to control the disease. The silencing of susceptibility genes through stable or transient transformation via genome editing or RNA interference (RNAi) is gaining importance [13,14]. For instance, RNAi has been successfully applied against *P. viticola* in greenhouse conditions, by targeting the *VviLBDIf7* gene encoding a LOB (LATERAL ORGAN BOUNDARIES) domain-containing (LBD) protein that belongs to plant LOB family of transcription factors [15]. To establish if the candidate resistance/susceptibility genes involved in N metabolism are truly modulated by the plant upon pathogen interaction and not by a differential presence of nitrogen in the plants, it is necessary to evaluate their basal expression under different N conditions and in absence of the pathogen. 

In this study, performed under controlled conditions, we aimed to assess the effect of N on both the pathogen development and the modulation of plant candidate resistance/susceptibility genes. To this purpose, we subjected grapevine cuttings to different levels of nitrogen fertilization for evaluating: (i) leaf nitrogen and chlorophyll content; (ii) *P. viticola* development and downy mildew severity; and (iii) the expression of the candidate resistance/susceptibility plant genes (*VvNRT3.1*, *VvGS* and *VvAS1*).

## 2. Results

Grapevine plants were grown in hydroponic conditions in a greenhouse and treated with different Ca(NO_3_)_2_ concentrations (0, 0.05, 1, 2, and 5 mM) after a period of N starvation. The optimal (control) concentration is 2 mM Ca(NO_3_)_2_. Leaf samples were collected at three timepoints (T1 = 7 days after the treatment; T2 = 17 days after the treatment; and T3 = 27 days after the treatment) for evaluating the N and chlorophyll content, the pathogen infection, and the expression of the target genes.

### 2.1. Nitrogen Content

Nitrogen quantification (N%) was evaluated at each timepoint (T1, T2 and T3) and for every Ca(NO_3_)_2_ treatment (0, 0.05, 1, 2 and 5 mM). The samples of control treatment (2 mM) showed the highest N percentage (3.45% ± 0.18) at all timepoints, while the samples treated with 0 (N = 2.9% ± 0.18) and 0.05 (N = 2.9% ± 0.23) mM Ca(NO_3_)_2_ showed the lowest concentrations (Figure 1A). Statistically significant differences with 2 mM samples (3.5 < F < 7.1; df = 4–10; *p* < 0.048) were found at T1 for 0.05 mM samples, at T2 for 0 mM and at T3 for 0 and 0.05 Ca(NO_3_)_2_ samples (Figure 1A). The leaves collected from the plants treated with 1 and 5 mM Ca(NO_3_)_2_) did not differ from the 2 mM leaves for the N content. Timing did not influence leaf N content apart from the unfertilized treatment, where N percentage significantly decreased from T1 to T2 and T3 (F = 8.84; df = 2–6; *p* = 0.016).

### 2.2. Photosynthetic Activity of the Plants

Leaf photosynthetic performance was evaluated by using an imaging chlorophyll fluorometer on the samples collected at T3 from the plants treated with the different Ca(NO_3_)_2_ concentrations. The average maximum quantum yield (MQY) values are reported in Table 1. Compared to the 2 mM treatment, the leaves collected from plants treated with lower or higher nitrogen concentrations showed a wider distribution towards lower MQY values (Figure 2A). The effect was particularly marked for the absence of fertilization. Significant differences among the different concentrations were found (KW = 43.276, df = 4, *p* < 0.001). Without acclimatization to darkness (t1), the highest average photosynthetic performance (0.75) was detected for the 2 mM treatment (Table 1). Fluorescence significantly decreased at increasing (5 mM) and decreasing (0.05 mM) N concentrations, reaching the lowest values (0.68) at 0 mM treatment. The 2 mM concentration statistically differed for MQY values from 0, 0.5 and 5 mM but not from 1 mM. 

After 15 min of darkness acclimatization (t15), photosynthetic performance did not show statistically significant differences between samples at 0.05, 1, 2, and 5 mM treatments (2 mM mean value = 0.39). Photosynthetic performance decreased at 0 mM treatment, showing statistically significant values (mean value = 0.3) compared to the 2 mM treatment (Table 1). The three highest Ca(NO_3_)_2_ concentrations (1, 2 and 5 mM) showed the highest chlorophyll content at both timepoints (Table 1).

### 2.3. Pathogen Development and Disease Severity

Leaf discs were cut from the sampled leaves and experimentally inoculated with *P. viticola* to quantify the sporulated area (SA, expressed in mm^2^), the percentage of sporulated area (PSA, expressed in percentage of sporulated leaf disc), and the sporangia produced by the pathogen (S, expressed as sporangia mm^–2^).

A significant positive correlation was found between N (%) and SA (mm^2^) (Kendall tau = 0.39; N = 45; *p* < 0.001; Spearman rho = 0.52; N = 45; *p* < 0.001) (Figure 1B).

The gamma regression model provided a comprehensive picture (McFadden’s pseudo R^2^ = 0.18) of the sporulated area (SA) outcome under different nitrogen concentrations while considering the different experimental inoculations. The ANOVA like test found significant differences in the interaction of predictors (F-value = 34.51; df = 8; *p* < 0.001) and in different Ca(NO_3_)_2_ concentrations (F-value = 42.01; df = 4; *p* < 0.001), whereas no differences among timepoints (F-value = 1.05; df = 2; *p* < 0.35) were found. For this reason, multiple comparisons on EMMs were restricted to the interaction term, to simultaneously visualize differences in disease severity among Ca(NO_3_)_2_ concentrations. The mean SA values showed little or no differences among Ca(NO_3_)_2_ concentrations at T1 and T2 (Table 2). The only slight difference among SA values was found at T1, when the 0.05 mM concentration showed a significant 2-fold reduction compared to the reference concentration (2 mM). The most important differences among SA values were found at the last timepoint (T3): the SA recorded in absence of nitrogen fertilization showed a significant 4-fold reduction compared with 2 mM Ca(NO_3_)_2_, with the other Ca(NO_3_)_2_ concentrations ranging in between these two. T3 was therefore chosen for further investigating the plant-pathogen interaction and collecting quantitative (disease severity and secondary inoculum quantification) and qualitative (microscopy) information of the infection process.

The distribution of disease severity values (PSA) showed a progressively positive skewing distribution at decreasing N concentrations (starting from 2 mM) (Figure 2B). In total absence of fertilization, the SA values were in most of the cases close to 0%. On the opposite, moving from 2 to 5 mM a tendency towards a negative skew can be noticed.

Compared to the reference, significantly low sporulation (KW = 35.02; df = 4; *p* < 0.0001) was found in absence of N fertilization (Table 2). The other Ca(NO_3_)_2_ ranged in between. According to Spearman’s Rho test (Rho: 0.70, *p* < 0.001), the sporangia production (S, expressed as sporangia mm^–2^) followed a positive and significant correlation with the sporulated area (mm^2^) (Figure 3).

The T3 inoculated leaf discs were stained with aniline blue to observe *P. viticola* structures (hyphae with haustoria developing in the intercellular spaces and sporangiophores emerging from the stomata) under an optical bright-field microscope [16]. Microscopy observations highlighted the greatest differences in *P. viticola* colonization between the reference nitrogen fertilization level (2 mM) and the absence of fertilization (0 mM). In detail, the pathogen development in leaves collected from plants treated with 2 mM Ca(NO_3_)_2_ followed a regular pattern, with the hyphae departing from the infected stomata, freely developing within the intercellular spaces and producing numerous haustoria for feeding (Figure 4A). On the opposite end, a total absence of nitrogen fertilization associated with a limited diffusion of the pathogen in the leaf tissues: the hyphae did not move far from the infected stomata and produced few haustoria (Figure 4B–D). Sporangiophores emerging from the stomata were sporadically seen (Figure 4B).

### 2.4. Expression of the Candidate Susceptibility/Resistance Genes

The relative gene expression of three genes (*VvNRT3.1*, *VvGS* and *VvAS1*) involved in N metabolism was determined per treatment (0, 0.05, 1, 2, and 5 mM Ca(NO_3_)_2_) and timepoint (T1, T2 and T3). Statistical analysis showed that a significantly higher abundance of *VvNRT3.1* transcripts was recorded in Pinot noir leaves treated with 1 mM Ca(NO_3_)_2_ at T3 (Figure 5A). This value was not statistically different from the values detected at 2 mM in all the three timepoints. At the reference condition (2 mM), not statistically different *VvNRT3.1* transcript abundances were detected among the three timepoints. The other combination of treatments and timepoints showed the lowest values. *VvGS* transcripts were abundant in leaves treated with 0 mM Ca(NO_3_)_2_ at T2, while the lowest values were detected in leaves treated with 5 mM Ca(NO_3_)_2_ (Figure 5B). The other combination of treatments and timepoints showed intermediate values. The abundance of *VvAS1* transcripts was not affected neither by treatment nor by time. No statistically significant differences were observed (Figure 5C).

## 3. Discussion

### 3.1. N Fertilization Influences Leaf N Content and the Photosynthesis

Nitrate (NO_3_^−^), the primary source of N for grapevine, is absorbed by roots and either stored in the vacuoles of the root cells or transported to the shoots through the xylem [5]. Before flowering, leaves and stems are the major N sinks. They are supplied initially by stored N reserves and successively by soil N [17]. The high nitrate concentration induced by fertilizer application can lead to N accumulation in the leaves [5], as observed in this study, where the treatment with 2 mM Ca(NO_3_)_2_ led to a significantly higher foliar N % than the absence or very limited (0.05 mM) fertilization. 

As a macronutrient, N plays a key role in plant metabolism, entering in the composition of key metabolites such as proteins, DNA, RNA, and chlorophyll [3]. N concentration in leaves is correlated with the chlorophyll index. Chlorophyll meter assays, such as chlorophyll fluorescence, reflect the intensity of leaf green color and are correlated with leaf chlorophyll and N concentrations [3]. Thus, chlorophyll fluorescence can be used to diagnose plant N status [8], as observed in our experimental conditions, where the treatment with optimal N fertilization (2 mM Ca(NO_3_)_2_) showed MQY values higher than the ones with non-optimal fertilization (0, 0.5 and 5 mM Ca(NO_3_)_2_).

### 3.2. Pathogen Development and Disease Severity Are Reduced in Absence of N Fertilization

Compared to the reference concentration (2 mM), the treatments with 1 and 5 mM Ca(NO_3_)_2_ induced statistically analogous SA at all timepoints, indicating that halving or increasing by 2.5 times the optimal N supply to the plant did not influence the ability of the pathogen to cause disease. Conversely, the plants that were subjected to no fertilization or to a 40 times reduction in the optimal N value registered the least disease severity values at T1 (for 0.05 mM concentration) and T3 (for no fertilization), indicating that only a substantial or complete lack in N supply reduces the ability of *P. viticola* to cause disease. This result confirms that the greatest responses to N fertilization are generally observed between deficiency and physiological sufficiency [2]. The results achieved at T3 are the most indicative of plants response to the different N treatments: at this timepoint, we can observe the greatest variability in the plant response to the pathogen, with SA values that are significantly different and PSA distributions that show a progressive decrease at decreasing nitrate supply (from 2 to 0 mM). This could be linked to the contemporaneous reduction in leaf N content that was observed in the 0 mM samples starting from T2 but was particularly marked at T3. The positive linear correlation observed between the SA and N content of the samples suggests that the disease severity is increased by N fertilization. Leaf nitrogen content, indeed, was already reported as one of the most important parameters influencing the diseased leaf area in field studies [8] and high N fertilization has been associated with increased susceptibility to *P. viticola* in a previous greenhouse experiment [18]. An analogous situation was already observed for other biotrophs such as the rust agents *Puccinia striiformis* f.sp. *tritici* (the yellow rust of winter wheat) [19], *Puccinia coronata* f. sp. *avenae* (the crown oat rust) [20] and the wheat powdery mildew agent *Blumeria graminis* f. sp. *tritici* [21]. 

The reasons behind the reduced susceptibility of plant to *P. viticola* at the low N content might be explained by the biotrophic nature of *P. viticola*. Most oomycetes use the major nitrogen sources found in planta (amino acids, ammonium, and nitrate), but the biotrophs have a greater reliance on the host because of their reduced metabolic capabilities [22]. Recently, it has been demonstrated that, like other obligate biotrophs [23], *P. viticola* lost genes encoding nitrate and nitrite reductase enzymes, becoming totally dependent from its host for acquiring nitrogen in its reduced form [24]. In general, nitrogen-deficient plants are not able to provide the nutrients that are necessary for the obligate pathogens [2]. This is testified by the poor vegetative growth of *P. viticola* observed at the microscope in leaf samples collected from 0 mM Ca(NO_3_)_2_ treated plants. Once penetrated through the stomata, the pathogen differentiated very short hyphae with few haustoria and sporadic sporangiophores; an indication that leaf tissue conditions were not favorable for its growth and asexual reproduction, as confirmed by the low sporangia production.

One of the interesting outcomes of the positive linear correlation obtained between the quantification of sporangia and sporulated area (SA) is that image analysis can be used for accurately estimating the sporangia production. This can help to reduce the amount of work needed for the phenotypic evaluation of the interaction between *P. viticola* and grapevine, which is time consuming and a limiting factor in the research of several features of downy mildew [25,26]. Quantifying the sporangia produced by *P. viticola* is important for biological and epidemiological reasons, because it is a fitness component and it can be used to estimate the inoculum produced by the pathogen to infect new plants and tissues [27,28,29,30]. This type of analysis is time consuming, since it must be performed at the microscope after detaching the sporangia from the leaves. Image analysis, on the contrary, is quite fast and simple because it is based on a picture of the sporulating leaf or leaf disc that is processed with a common image analysis software, it can be automated, and it provides more precise and reliable data than the traditional visual assessment methods [31].

### 3.3. The Expression of the Candidate Susceptibility/Resistance Genes Is Not Influenced by N Fertilization

Nitrogen is an essential nutrient for plants [32] that can acquire both nitrate (NO_3_^−^) or ammonium (NH_4_^+^) mineral forms [33]. Its uptake and allocation are mediated by carrier proteins showing different affinities (low- and high-affinity transporters, NRT) for N (NO_3_^−^ and NH_4_^+^) forms [34]. At nitrate concentrations above 1 mM, N acquisition is mainly operated by Low-Affinity Transport System (LATS), while when the nitrate concentration is below 1 mM, N acquisition is mainly operated by High-Affinity Transport System (HATS) [35]. *NRT3.1* is considered to be a high-affinity transporter proved to be responsive to nitrate induction in *Arabidopsis thaliana* [36]. In this work, *VvNRT3.1* acted as a low-affinity transporter. It was expressed 27 days after treatment (T3) with 1 mM nitrate, suggesting its involvement in nitrate transport at low nitrate supply level [11,37]. In higher plants, glutamine synthetase catalyzes the incorporation of ammonia into glutamate to generate glutamine, being a major enzyme responsible for the assimilation of ammonium absorbed from the growth medium or generated by NO_3_^−^ reduction [38]. GS activity seemed to not or only slightly be affected by N form [39]. In our experimental conditions, *VvGS* was slightly active at low nitrate concentration (0 and 0.05 mM), 7 and 17 days (T1 and T2) after the beginning of treatment, meaning that the plant is perceiving a non-optimal N condition [11,37,40,41]. Asparagine synthetase catalyzes the synthesis of asparagine from aspartate, playing a role in both N assimilation and remobilization [42,43]. The expression of gene encoding for this enzyme has been found to be affected by both form and content. In *A. thaliana*, the expression level of *AtAS2* gene is induced by NH_4_^+^, while *AS* genes are induced by NO_3_^−^ in *Phaseolus vulgaris* and soybean [44,45,46], but the expression level of *VvAS1* in leaves of cv Pinot noir seems not to be affected by NO_3_^−^ treatments [37,47,48].

All of these genes are induced in response to pathogens. In *A. thaliana*, two *NRT* genes are involved in the response to *Pseudomonas syringae* and to *Erwinia amylovora* [49]. In grapevine, *VvNRT3.1* seems to be involved in the response to *P. viticola* infection. It was upregulated in infected leaves of cv Bianca, and was downregulated in infected leaves of cv Pinot noir [37]. *GS1* was upregulated in tomato cultivars resistant to *Botrytis cinerea* [40] and the expression *GS1* has been shown to be induced in *A. thaliana* upon fungal infections [41]. The *GS1* involvement in plant stress response has been observed also in grapevine. *VvGS1* was differentially expressed in resistant cultivars infected with *P. viticola* [37]. In non-optimal N conditions, it can be hypothesized that the activation of *VvNRT3.1* and *VvGS1* could lead the plant to show a less susceptible phenotype. *AS1* is required not only for plant nitrogen assimilation, but also for defense response to microbial pathogens [47,48]. *VvAS1* has been identified to be involved in the response of grapevine to *P. viticola* infection. Its expression was down-regulated in cv Bianca and upregulated in cv Pinot noir infected leaves [37]. The activation of *VvAS1* in infected leaves of cv Pinot noir may play a role in promoting *P. viticola* pathogenesis by providing a rich source of N for the pathogen. The putative role in pathogenicity and its steady expression in optimal and non-optimal N conditions make *VvAS1* a good candidate susceptibility gene. 

## 4. Materials and Methods

### 4.1. Plant Material, Growth Conditions and Sampling

The experimental activities were conducted in the greenhouse of Department of Agricultural and Environmental Sciences (University of Milan, Milan, Italy), with a 12 h-photoperiod and 24 °C. Thirty vine cuttings of cv Pinot noir grafted onto SO4 were put in 25-L aerated tanks filled with a hydroponic nutrient solution for shooting and growing. The tanks were initially filled with 20 L of Hoagland solution for hydroponic culture: 2 mM Ca(NO_3_)_2_, 0.75 mM K_2_SO_4_, 0.65 mM MgSO_4_, 0.5 mM KH_2_PO_4_, 0.1 mM FeNaEDTA, 0.005 mM H_3_BO_3_, 0.001 mM MnSO_4_, 0.0005 mM CuSO_4_, 0.0005 mM ZnSO_4_, 0.00005 mM (NH_4_)_6_Mo_7_O_24_, pH 6.1 [50]. After 30 days, the developed plants were grown in N starvation (Hoagland solution without N) for 15 days and then divided into five groups. Each group was exposed to Hoagland nutrient solutions containing different Ca(NO_3_)_2_ concentrations: 0, 0.05, 1, 2 (standard reference concentration) and 5 mM. All of the solutions, prepared with distilled water, were changed weekly with fresh ones. The starvation period was necessary to deplete the N reserves that are located in grapevine roots [3] and make the treatments with different N concentrations effective. Starting from T0 (the first day of exposure to the five different Hoagland solutions), three leaf samplings (T1, T2 and T3) were performed: T1 leaves were collected 7 days after the treatment, while T2 and T3 at 17 and 27 days after treatment, respectively. Leaves were used to: (i) quantify N amount; (ii) evaluate photosynthetic performance (only on T3 samples); (iii) inoculate *P. viticola* and determine disease severity; (iv) evaluate expression of genes related to N metabolism. Five leaves were randomly collected from three different plants per treatment at each sampling time (Figure 6). From these leaves, 12 leaf discs were immediately collected with a cork borer (Ø 1.5 cm) and used for pathogen inoculation and photosynthetic performance evaluation (Figure 6A,B). The remaining leaf tissues of three leaves were stored at room temperature until N quantification (Figure 6C), while the tissues of the other two leaves were then immediately frozen with liquid nitrogen and kept at −80 °C until RNA extraction (Figure 6D).

### 4.2. Nitrogen Quantification

After sampling, leaves were dried (60 °C, 2 days) to a stable dry weight. Dried leaves were ground using the Retsch MM300 tissuelyser (Gemini B.V., Apeldoom The Netherlands). Levels of leaf N were estimated using an elemental analyzer NA 1500 series 2 NC (Carlo Erba, Italy), starting from 3 mg of each ground leaf sample [51].

### 4.3. Photosynthetic Performance Analysis

At T3, chlorophyll fluorescence was measured to evaluate the photosynthetic performance associated with excised leaf discs through an imaging chlorophyll fluorometer Imaging PAM (Walz, Effeltrich, Germany). Maximum quantum yield (MQY) and effective quantum yield of photosystem II were measured and imaging data analyzed using the ImagingWinGigE (Walz, Germany) software to retrieve the MQY mean value associated with each leaf disc [52,53,54]. Two measurements per sample were performed: the first measure (t1) was done without acclimatization to darkness, while the second (t15) was performed after 15 min of acclimatization to darkness.

### 4.4. Pathogen Infection

The leaf discs were placed with their abaxial surface up in a Petri dish (9 cm diameter) containing moistened filter paper underneath. Each leaf disc was inoculated with five drops (5 μL) of a *P. viticola* sporangia suspension (5 × 10^5^ sporangia mL^−1^) obtained by washing previously infected leaves with sterile distilled water. The *P. viticola* inoculum used was a mix of three different populations with different geographical origins (Oltrepò Pavese, Franciacorta and Friuli-Venezia Giulia) [16]. After the inoculation, plates were incubated for seven days in a growth chamber at 22 °C (18:06 h photoperiod) and dried with filter paper for 24 h post-inoculation. After 7 days, a picture was taken with a digital camera (Fujifilm Digital camera A160) for each plate, and the sporulated area (SA) of each leaf disc (mm^2^) was determined by manual segmentation of pixels [55,56] corresponding to sporulation using the GIMP 2.10 software (https://docs.gimp.org/2.10/en/, accessed on 5 June 2022). The disease severity was expressed as the percentage of sporulated area (PSA) over the total leaf disc area. For T3 samples only, sporangia were collected from the leaf discs in 1 mL of sterile distilled water-glycerol solution (80:20 *v*/*v*) and counted in KOVA chambers (KOVA International Inc., Garden Grove, CA, USA) to calculate the number of sporangia (S) produced by the pathogen per leaf unit (sporangia mm^–2^) [16]. The inoculated leaf discs were then fixed in Carnoy’s solution (1:3 glacial acetic acid:absolute ethanol), and stored at 4 °C until use, then the discs were cleared and stained with aniline blue [16]. The pathogen structures, visible in the blue inside the stained leaf discs, were observed at Zeiss Primo Vert optical microscope equipped with Primo Cam HD5 camera (Tiesselab, Milano, Italy). For maintaining continuity with the inoculum over time, the artificial *P. viticola* population used for infecting T1 leaves was propagated on fresh detached leaves obtained from potted plants of cv Pinot noir following the above described protocol.

### 4.5. Gene Expression Analysis

Total RNA was isolated from leaves, frozen in liquid nitrogen after sampling, and stored at −80 °C until their use. Leaves were ground in liquid nitrogen and 100 mg of ground tissue were used to perform RNA extraction, using Spectrum™ Plant Total RNA Kit (Sigma-Aldrich, Darmstadt, Germany), according to manufacturer’s instructions. Qubit^®^ 3.0 Fluorometer (Life Technologies, Carlsbad, CA, USA) was used for RNA quantification, using Qubit^®^ RNA HS Assay Kit (Life Technologies), while NanoDrop 8000 Spectrophotometer (Thermo Fisher Scientific, Waltham, MA, USA) was used to check RNA quality (260/230 and 260/280 ratios). Lithium-chloride treatment was not required, as no samples showed a 260/230 ratio lower than 1.8. Five hundred ng of total RNA were reverse transcribed with SuperScript^®^IV Reverse Transcriptase (Thermo Fisher Scientific), according to manufacturer’s instructions. Semi-quantitative real-time RT-PCR was carried out on QuantStudio^®^ 3 Real-Time PCR Systems (Thermo Fisher). Each reaction was carried out in a volume of 20 µL, using 10 µL of PowerUp™ SYBR™ Green Master Mix (Applied Biosystems Waltham, MA, USA), 500 nM of each primer, 4 µL of cDNA diluted 1:10 and water up to the final volume. Each reaction was performed in triplicate. The expression of three genes was evaluated: nitrate transporter (*VvNRT3.1*), glutamine synthetase (*VvGS*) and asparagine synthetase (*VvAS1*) (Table 3). Primers were designed on the available sequences using the Primer3 Plus software (Untergasser et al. 2007). Sequence primers are reported in Table 3. Ubiquitin (Fujita et al. 2007) and glyceraldehyde-3-phosphate dehydrogenase [57] genes were used as references for data normalization. The expression of each gene was calculated following the 2^−ΔΔCt^ method [58].

### 4.6. Statistical Analysis

Statistical analyses of nitrogen quantification were performed using the R package agricolae (version 3.6.0; [59]) via ANOVA test and Fisher’s least significant difference (LSD) post-hoc test with Bonferroni’s correction. Statistical analysis of photosynthetic performances was performed using the R package agricolae via Kruskal-Wallis H test and Fisher’s least significant difference (LSD) post-hoc test with Bonferroni’s correction.

The continuous outcome provided by the leaf disc surface covered by sporulation (mm^2^) can be assumed as a sample of gamma distribution [60,61,62]. For this reason, a gamma family multiple generalized linear model (GLM) was fitted [gy = β_0_ + β_ij_X_ij_ + β_1_X_i_X_j_ where: β_0_ is the model intercept; β_ij_ the parameter’s intercept and β_1_ the interaction term’s intercept, whereas g(•) represent the link function (log)] by using glm function in R stats. In detail, the disease severity data (y) were transformed as y + 1 to include null values, whereas the X_ij_ parameter was represented by two categorical factors: the i-th experimental timepoint (T1, T2 and T3) and the j-th nitrogen concentration (0, 0.05, 1, 2 and 5 mM Ca(NO_3_)_2_), respectively. The X_i_X_j_ parameter represented the interaction term. Estimated Marginal Means (EMMs) were retrieved from the fitted model via emmeans and multiple pairwise comparisons (with Sidak’s correction) among estimated EMMs were calculated through pairs function. In addition, to account for the overall effect of independent variables, an ANOVA-like type III test was performed through joint_test in emmeans package (https://CRAN.R-project.org/package=emmeans, accessed on 5 June 2022). 

A Spearman’s Rho correlation test (cor.test in R stats) was employed to measure the strength of association between the disease severity (PSA), and the number of sporangia per leaf unit (sporangia mm^−2^). The distribution of SA values at T1, T2 and T3 was represented with violin plots, i.e., a combination of a box plot and a density plot which shows peaks in the data [63]. In detail, violin plots show inside the box-plot distribution of the data, with the median, quartile, and outside the distribution shape of the data, where the width of the box is proportional to the estimated density: wider and skinnier sections of the violin plot represent, respectively, higher and lower estimated densities.

To account for differences in the MQY values recorded at different Ca(NO_3_)_2_ concentrations, a Kruskal Wallis test followed by a Fisher’s least significant difference post-hoc test using kruskal in agricolae package was performed [59]. All of the statistical analyses were performed using R (https://www.r-project.org/, accessed on 5 June 2022) in R studio 9.1 (http://www.rstudio.com, accessed on 5 June 2022).

The 2^−ΔΔCt^ values were subjected to Levene’s test to assess the homogeneity of variance in R software. LSD test was performed in R to evaluate differences in the expression of *VvNRT3.1*, *VvGS* and *VvAS1* genes across all the different N conditions.

## 5. Conclusions

The results highlighted that only a very low level of nitrogen availability could reduce the downy mildew severity, while small increases or decreases of N supply compared to the standard reference Ca(NO_3_)_2_ concentration (2 mM) did not show any sensible effect on the management of the disease. *VvNRT3.1*, *VvGS* and *VvAS1* are good candidate resistance and susceptibility genes, because their expression was not influenced by nitrogen nutrition. Further analyses are needed to assess the effect of other nitrogen nutritional levels on the expression of these genes, and to evaluate the role of the proteins encoded by the candidate resistance/susceptibility genes in the plant-pathogen interaction. However, the present achievements strongly indicate that these genes can be exploited for the selection and development of disease management tools. In particular, susceptibility genes could be transiently knocked down by exploiting target specific dsRNA to inhibit the pathogen development, or stably knocked out by genome editing, while resistance genes could be exploited as markers of the plant response to the pathogen.

## Figures and Tables

**Figure 1 plants-12-00263-f001:**
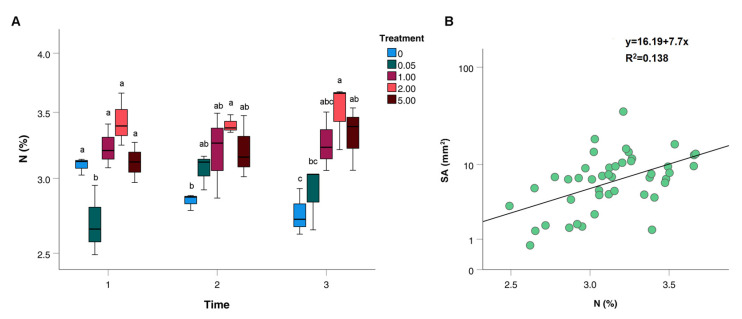
(**A**) Box-plot distribution of nitrogen (N) content (%) at different Ca(NO_3_)_2_ concentrations. Different letters indicate significant differences (*p* < 0.05) in N (%) among treatments within timepoint. (**B**) Scatter plot showing the positive linear correlation existing between N content (%) and sporulated area (SA; mm^2^).

**Figure 2 plants-12-00263-f002:**
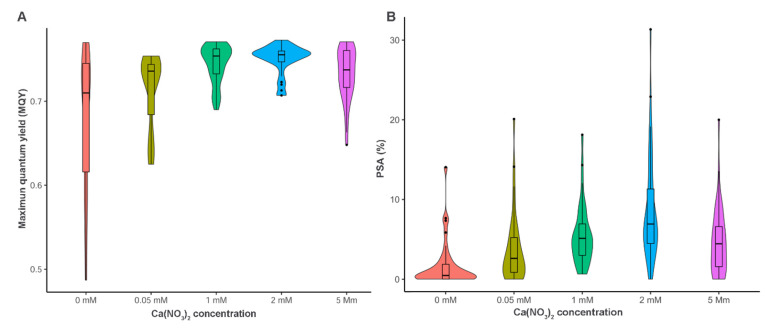
Violin plot distribution of (**A**) maximum quantum yield (MQY) without acclimatization and (**B**) disease severity (PSA) recorded at T3 in the leaf discs sampled from plants treated with different Ca(NO_3_)_2_ concentrations (0, 0.05, 1, 2 and 5 mM). The width of the plot is proportional to the estimated density.

**Figure 3 plants-12-00263-f003:**
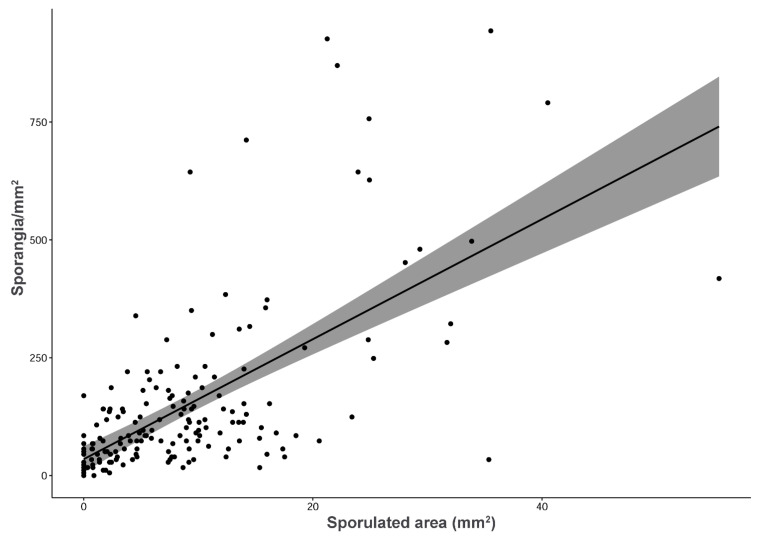
Scatter plot showing the positive linear correlation existing between sporangia production (sporangia mm^–2^) and sporulated area (mm^2^). Grey shade represents confidence interval.

**Figure 4 plants-12-00263-f004:**
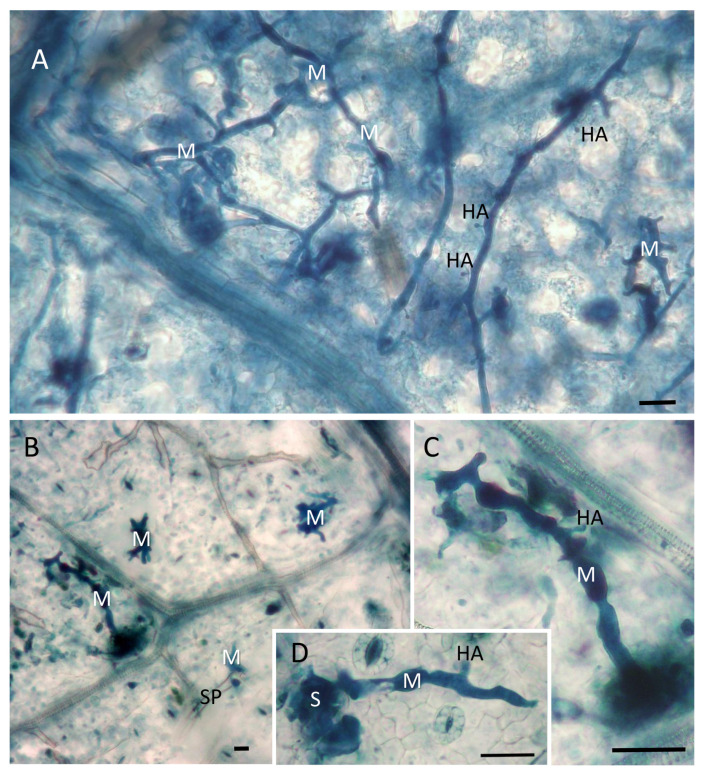
*P. viticola* structures inside leaves sampled at T3 from plants that were fertilized with 2 mM nitrogen (**A**) or not fertilized (**B**–**D**) and observed at 7 dpi (day post inoculation). (**A**) Mycelium with haustoria regularly elongating and branching in the mesophyll cells. (**B**) Short portions of mycelium departing from infected stomata with a sporangiophore. (**C**,**D**) Short hypha with a clearly visible haustorium. S = stoma; M = mycelium; HA = haustorium; SP = sporangiophore. Scale bar = 50 µm.

**Figure 5 plants-12-00263-f005:**
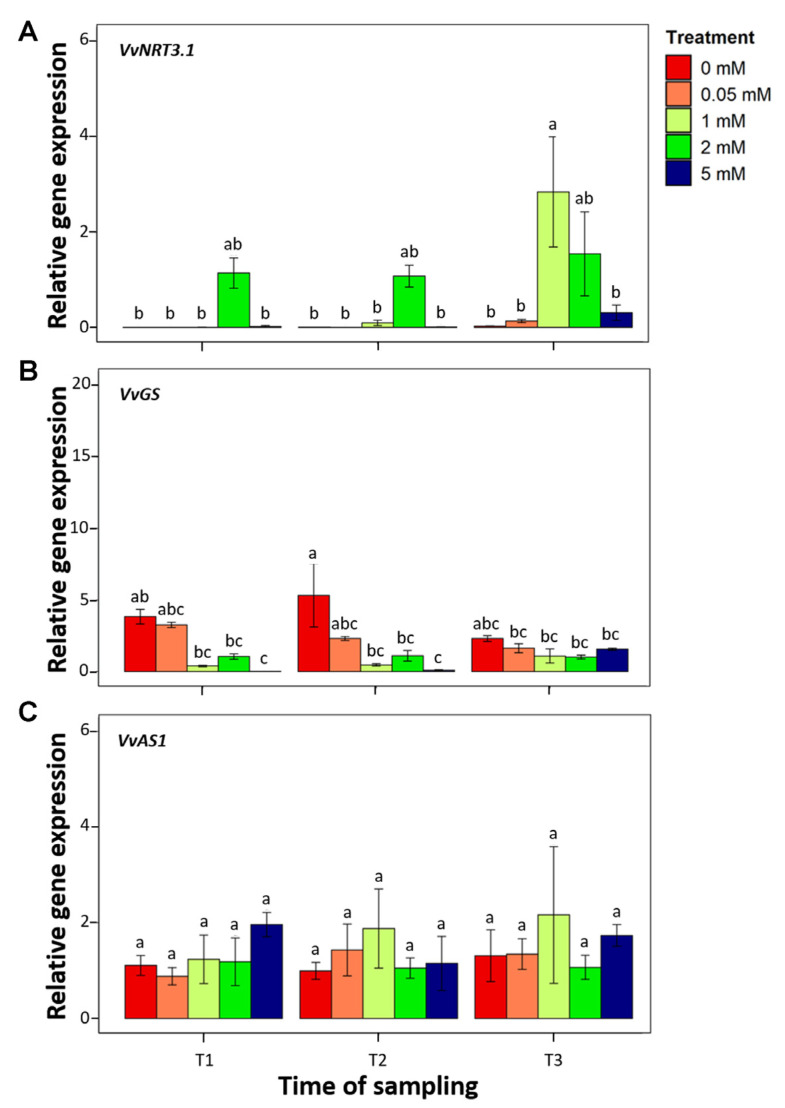
Relative expression level (2^−ΔΔCt^ method) of genes involved in the N metabolism detected in leaves of Pinot noir plants grown under different N concentrations (0, 0.05, 1, 2 and 5 mM Ca(NO_3_)_2_) at different timepoints (T1, T2 and T3). (**A**) Nitrate transporter (*VvNRT3.1*). (**B**) Glutamine synthetase (*VvGS*). (**C**) Asparagine synthetase (*VvAS1*). Bars represent the standard deviation. Different letters indicate statistically significant differences.

**Figure 6 plants-12-00263-f006:**
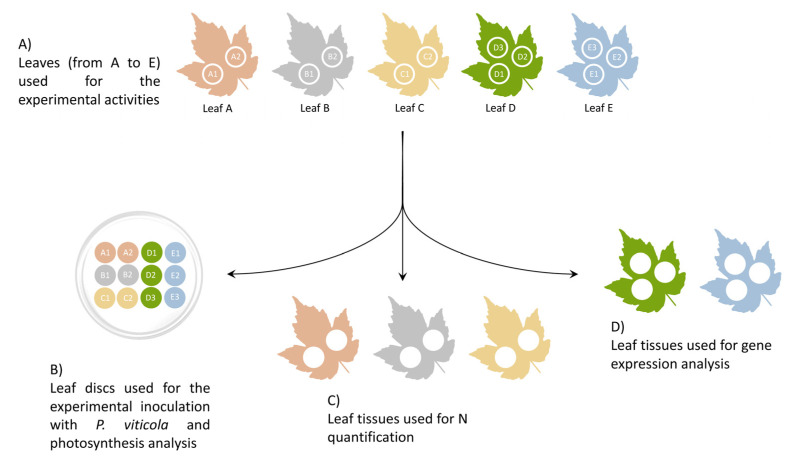
Scheme of the utilization of the leaves. (**A**) Five leaves (A, B, C, D, and E) were sampled for each treatment. Two leaf discs (A1-2, B1-2 and C1-2) were cut from leaves A–C, while three leaf discs (D1-3 and E1-3) were sampled from leaves D and E. (**B**) The leaf discs were placed, lower surface upwards, in a Petri dish for experimental inoculation with *P. viticola*. The leaf tissues remaining from leaves A–C were used for N quantification (**C**) while those remaining from leaves D and E were used for gene expression analysis (**D**).

**Table 1 plants-12-00263-t001:** Mean measured MQY (maximum quantum yield) values ± standard deviations at each Ca(NO_3_)_2_ concentration before (t1) and after 15 min of dark acclimatization (t15) at T3. Means sharing a letter are not significantly different.

Ca(NO_3_)_2_ Concentration (mM)	t1	t15
0	0.68 ± 0.08 a	0.30 ± 0.08 a
0.05	0.71 ± 0.04 ab	0.38 ± 0.07 b
1	0.74 ± 0.02 cd	0.37 ± 0.05 b
2	0.75 ± 0.02 d	0.38 ± 0.04 b
5	0.73 ± 0.03 bc	0.39 ± 0.05 b

**Table 2 plants-12-00263-t002:** Mean sporulated area (SA, mm^2^) and sporangia production (S = sporangia mm^–2^) ± standard deviation at each Ca(NO_3_)_2_ concentrations and timepoint. Different letters indicate significant differences among estimated marginal means (EMMs) of SA or mean S values (*p* < 0.05).

Ca(NO_3_)_2_ Concentration (mM)	SA (mm^2^)			S (Sporangia mm^–2^)
	T1	T2	T3	T3
0	9.9 ± 6.37 cd	5.01 ± 4.96 abc	2.99 ± 5.14 a	56.97 ± 54.02 a
0.05	3.94 ± 3.35 ab	8.84 ± 6.58 c	6.6 ± 7.83 abc	128.84 ± 121.99 ab
1	7.59 ± 3.76 bc	7.93 ± 6.08 bc	9.64 ± 6.68 cd	168.55 ± 160.54 bc
2	9.00 ± 6.03 cd	7.48 ± 5.3 bc	15.37 ± 11.73 d	244.04 ± 230.84 c
5	8.12 ± 5.28 bc	7.61 ± 5.71 bc	8.42 ± 7.62 bcd	103.58 ± 127.62 ab

**Table 3 plants-12-00263-t003:** List of genes analyzed in this work. Gene ID, protein name and primers are reported.

Gene	Gene ID	Protein Name	Primers (5′ → 3′)
*VvNRT3.1*	XM_002279825.3	high-affinity nitrate transporter 3.1	F: TTCCAACAATTCCGTAGAGTGG R: AGTTACGGTTTCTCTAATCACC
*VvGS*	XM_002274103.2	glutamine synthetase leaf isozyme	F: GCTGATCTCCAGAACATCAACC R: CGGCAGTTGCGCTGGGCCAGTG
*VvAS1*	NM_001281237.1	asparagine synthetase	F: GCAATGGATATTGACCCTGAGTG R: AAAAGCCCTCCTGAGAACCC

## Data Availability

All research data are reported in the article.

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
