# Peer review of "Influence of Nitrogen on Grapevine Susceptibility to Downy Mildew"

_plants, 2023, doi:10.3390/plants12020263_

Round 1

Reviewer 1 Report

The manuscript entitled “Influence of nitrogen on grapevine susceptibility to downy mildew” provide a new insight into grapevine susceptibility to Plasmopara viticola based on the mRNA transcription level. Numerous studies confirm the low correlation between transcript and protein levels. Thus, the study should be performed on the protein level to ensure the exact importance of three mentioned proteins. The manuscript is well written, and all methods are conducted according to high scientific standards. However, before publication, some minor changes should be made. In tables 1 and 2, the number of decimal places must be adjusted. Lines 94 and 163, please change 2mM to 2 mM.

Author Response

Dear Reviewer, on behalf of the authors I wish to thank you for the precious comments on the manuscript. We made the following modifications:

  • We added the following phrase “Further analyses are needed to assess the effect of other nitrogen nutritional levels on the expression of these genes and to evaluate the role of the proteins encoded by the candidate resistance/susceptibility genes in the plant-pathogen interaction” in the Conclusion section;
  • Two decimals were inserted in Tables 1 and 2;
  • “2mM” was changed into “2 mM” at lines 94 and 163.

Best regards,

Silvia Toffolatti

Reviewer 2 Report

The authors have compiled a  very interesting topic  "Influence of nitrogen on grapevine susceptibility to downy mildew" . Although experiment is well designed and clearly presented ,but article need slightly modifications in english. I am mentioning herewith some points:

Line55-58 Even if nitrogen fertilization is known for influencing the disease severity of P. viticola in 56 the open field, no information is available on the effect of nitrogen fertilization on the 57 plant infection process and on the mechanisms that are behind nitrogen fertilization and 58 plant-pathogen interaction

Line84-performed in controlled conditions in greenhouse

Fig 1.b  increase the text size

Line-Leaves photosynthetic performance was evaluated at T3 overall the different  Ca(NO3)2 concentrations, using an imaging chlorophyll fluorometer- check and reframe the sentence

Add any reference- 446-450

Line 460, 470-  why () ?

Author Response

Dear Reviewer, on behalf of the authors I wish to thank you for the precious comments on the manuscript. We made the following modifications:

  1. The phrase at lines 55-58 was changed into “Even if nitrogen fertilization is known for influencing the disease severity of viticola in the open field, no information is available on the specific mechanisms underlying its role in the plant-pathogen interaction.”;
  2. “In this study, performed in controlled conditions in greenhouse” was changed into “In this study, performed under controlled conditions”;
  3. The text size of the equation was increased in Fig. 1b;
  4. “Leaves photosynthetic performance was evaluated at T3 overall the different Ca(NO3)2 concentrations, using an imaging chlorophyll fluorometer” was changed into “Leaves photosynthetic performance was evaluated by using an imaging chlorophyll fluorometer on the samples collected at T3 from the plants treated with the different Ca(NO3)2 concentrations”;
  5. The following citations were added for the gamma model:
    • Segarra, J.; Jeger, M.J.; Bosch, F. Van Den Epidemic Dynamics and Patterns of Plant Diseases. Phytopathology 2001, 91, 1001–1010, doi:10.1094/PHYTO.2001.91.10.1001.
    • Thom, H. A note on the gamma distribution. Mon. Weather Rev. 1958, 86, 117–122, doi:10.1016/0016-0032(94)90228-3.
    • van den Bosch, F.; Zadoks, J.; Metz, J. Focus Expansion in Plant Disease. II: Realistic Parameter-Sparse Models. Phytopathology 1988, 78, 59–64, doi:10.1094/phyto-78-59.
  6. The empty brackets are just a part indicating the function in R statistical package;
  7. The manuscript was revised for the English language (please, see modifications in the manuscript file).

Best regards,

Silvia Toffolatti and Gabriella De Lorenzis